# SCALING PHYSICS-INFORMED HARD CONSTRAINTS WITH MIXTURE-OF-EXPERTS

**Nithin Chalapathi, Yiheng Du, Aditi S. Krishnapriyan**
`{nithinc, yihengdu, aditik1}@berkeley.edu`
University of California, Berkeley

## ABSTRACT

Imposing known physical constraints, such as conservation laws, during neural network training introduces an inductive bias that can improve accuracy, reliability, convergence, and data efficiency for modeling physical dynamics. While such constraints can be softly imposed via loss function penalties, recent advancements in differentiable physics and optimization improve performance by incorporating PDE-constrained optimization as individual layers in neural networks. This enables a stricter adherence to physical constraints. However, imposing hard constraints significantly increases computational and memory costs, especially for complex dynamical systems. This is because it requires solving an optimization problem over a large number of points in a mesh, representing spatial and temporal discretizations, which greatly increases the complexity of the constraint. To address this challenge, we develop a scalable approach to enforce hard physical constraints using Mixture-of-Experts (MoE), which can be used with any neural network architecture. Our approach imposes the constraint over smaller decomposed domains, each of which is solved by an "expert" through differentiable optimization. During training, each expert independently performs a localized backpropagation step by leveraging the implicit function theorem; the independence of each expert allows for parallelization across multiple GPUs. Compared to standard differentiable optimization, our scalable approach achieves greater accuracy in the neural PDE solver setting for predicting the dynamics of challenging non-linear systems. We also improve training stability and require significantly less computation time during both training and inference stages.

## 1 INTRODUCTION

Many problems necessitate modeling the physical world, which is governed by a set of established physical laws. For example, conservation laws such as conservation of mass and conservation of momentum are integral to the understanding and modeling of fluid dynamics, heat transfer, chemical reaction networks, and other related areas (Herron & Foster, 2008). Recently, machine learning (ML) approaches, and particularly neural networks (NNs), have shown promise in addressing problems in these areas.

The consistency of these physical laws means that they can provide a strong supervision signal for NNs and act as inductive biases, rather than relying on unstructured data. A common approach to incorporate physical laws into NN training is through a penalty term in the loss function, essentially acting as a soft constraint. However, this approach has drawbacks, as empirical evidence suggests that soft constraints can make the optimization problem difficult, leading to convergence issues (Krishnapriyan et al., 2021; Wang et al., 2021). Additionally, at inference time, soft constraints offer no guarantee of constraint enforcement, posing challenges for reliability and accuracy.

Alternatively, given the unchanging nature of these physical laws, there lies potential to create improved constraint enforcement mechanisms during NN training. This can be particularly useful when there is limited training data available, or a demand for heightened reliability. This notion segues into the broader concept of a differentiable physics approach, where differentiation through

physical simulations ensures strict adherence to the underlying physical dynamics (Amos & Kolter, 2017; Qiao et al., 2020; Ramsundar et al., 2021; Kotary et al., 2021).

To solve a physical problem, typical practice in scientific computing is to use a mesh to represent the spatiotemporal domain. This mesh is discretized, and physical laws can be enforced on the points within the discretization. This can be considered an equality constrained optimization problem, and there are lines of work focusing on incorporating such problems as individual layers in larger end-to-end trainable NNs (Négiar et al., 2023; Amos & Kolter, 2017). The advantages of this approach include stricter enforcement of constraints during both training and inference time, which can lead to greater accuracy than the aforementioned soft constraint settings.

However, these approaches, aimed at enforcing constraints more precisely ("hard" constraints), also face a number of challenges. Backpropagating through constraints over large meshes is a highly non-linear problem that grows in dimensionality with respect to the mesh and NN model sizes. This means that it can be both computationally and memory intensive. This scenario epitomizes an inherent trade-off between how well the constraint is enforced, and the time and space complexity of enforcing the constraint.

To address these challenges, we propose a mixture-of-experts formulation to enforce equality constraints of physical laws over a spatiotemporal domain. We focus on systems whose dynamics are governed by partial differential equations (PDEs), and impose a set of scalable physics-informed hard constraints. To illustrate this, we view our framework through the neural PDE solver setting, where the goal is to learn a differential operator that models the solution to a system of PDEs. Our approach imposes the constraint over smaller decomposed domains, each of which is solved by an "expert" through differentiable optimization. Each expert independently performs a localized optimization, imposing the known physical priors over its domain. During backpropagation, we then compute gradients using the implicit function theorem locally on a per-expert basis. This allows us to parallelize both the forward and backward pass, and improve training stability.

Our main contributions are as follows:

- We introduce a physics-informed mixture-of-experts training framework (PI-HC-MoE), which offers a scalable approach for imposing hard physical constraints on neural networks by differentiating through physical dynamics represented via a constrained optimization problem. By localizing the constraint, we parallelize computation while reducing the complexity of the constraint, leading to more stable training.

- We instantiate PI-HC-MoE in the neural PDE solver setting, and demonstrate our approach on two challenging non-linear problems: diffusion-sorption and turbulent Navier-Stokes. Our approach yields significantly higher accuracy than soft penalty enforcement methods and standard hard-constrained differentiable optimization.

- Through our scalable approach, PI-HC-MoE exhibits substantial efficiency improvements when compared to standard differentiable optimization. PI-HC-MoE exhibits sub-linear scaling as the hard constraint is enforced on an increasing number of sampled points within the spatiotemporal domain. In contrast, the execution time for standard differentiable optimization significantly escalates with the expansion of sampled points.

- We release our code[1], built using JAX, to facilitate reproducibility and enable researchers to explore and extend the results.

## 2 RELATED WORK

**Constraints in Neural Networks.**   Prior works that impose constraints on NNs to enforce a prior inductive bias generally fall into two categories: soft and hard constraints. Soft constraints use penalty terms in the objective function to push the NN in a particular direction. For example, physics-informed neural networks (PINNs) (Raissi et al., 2019) use physical laws as a penalty term. Other examples include enforcing Lipschitz continuity (Miyato et al., 2018), Bellman optimally (Nikishin et al., 2022), turbulence (List et al., 2022), numerical surrogates (Pestourie et al., 2023), prolongation matrices (Huang et al., 2021), solver iterations (Kelly et al., 2020), spectral

---

[1]`https://github.com/ASK-Berkeley/physics-NNs-hard-constraints`

methods Du et al. (2023), and convexity over derivatives (Amos et al., 2017). We are primarily concerned with *hard constraints*; methods that, by construction, exactly enforce the constraint at train and test time. Hard constraints can be enforced in multiple ways, including neural conservation laws (Richter-Powell et al., 2022), PDE-CL (Négiar et al., 2023), BOON (Saad et al., 2023), constrained neural fields (Zhong et al., 2023), boundary graph neural networks (Mayr et al., 2023), PCL (Xu & Darve, 2022), constitutive laws (Ma et al., 2023), boundary conditions (Sukumar & Srivastava, 2022), characteristic layers Braga-Neto (2023), and inverse design (Lu et al., 2021).

**Differentiable Optimization.** An approach to impose a hard constraint is to use differentiable optimization to solve a system of equations. Differentiable optimization folds in a second optimization problem during the forward pass and uses the implicit function theorem (IFT) to compute gradients. OptNet (Amos & Kolter, 2017) provides one of the first formulations, which integrates linear quadratic programs into deep neural networks. DC3 (Donti et al., 2021) propose a new training and inference procedure involving two steps: completion to satisfy the constraint and correction to remedy any deviations. A related line of work are implicit neural layers (Blondel et al., 2022; Chen et al., 2018), which use IFT to replace the traditional autograd backprop procedure. Theseus (Pineda et al., 2022) and JaxOpt (Blondel et al., 2022) are two open-source libraries implementing common linear and non-linear least squares solvers on GPUs, and provide implicit differentiation capabilities. Our formulation is agnostic to the exact method, framework, or iterative solver used. We focus on non-linear least squares solvers because of their relevance to real-world problems, though many potential alternative solver choices exist (Berahas et al., 2021; Fang et al., 2024; Na et al., 2022).

**Differentiable Physics.** Related to differentiable optimization, differentiable physics (de Avila Belbute-Peres et al., 2018) embeds physical simulations within a NN training pipeline (Ramsundar et al., 2021). Qiao et al. (2020) leverage meshes to reformulate the linear complementary problem when simulating collisions. This line of work is useful within the context of reinforcement learning, where computing the gradients of a physics state simulator may be intractable. Differentiable fluid state simulators (Xian et al., 2023; Takahashi et al., 2021) are another example.

**Mixture-of-Experts.** MoE was popularized in natural language processing as a method to increase NN capacity while balancing computation (Shazeer et al., 2017). The key idea behind MoE is to conditionally route computation through "experts", smaller networks that run independently. MoE has been used in many settings, including vision (Ruiz et al., 2021) and GPT-3 (Brown et al., 2020).

## 3 METHODS

**General problem overview.** We consider PDEs of the form $\mathcal{F}_\phi(u) = \mathbf{0}$, where $u : \Omega \to \mathbb{R}$ is the solution and $\phi$ may represent a variety of parameters including boundary conditions, initial conditions, constant values (e.g., porosity of a medium), or varying parameters (e.g., density or mass). Here, $\Omega$ defines the spatiotemporal grid. We would like to learn a mapping $\phi \mapsto u_\theta$, where $u$ is parameterized by $\theta$ such that $\mathcal{F}_\phi(u_\theta) = \mathbf{0}$. This mapping can be learned by a neural network (NN). The *soft constraint* enforces $\mathcal{F}_\phi(u_\theta) = \mathbf{0}$ by using a penalty term in the loss function. That is, we can minimize $||\mathcal{F}_\phi(u_\theta)||_2^2$ as the loss function.

**Enforcing Physics-Informed Hard Constraints in Neural Networks.** We enforce physics-informed hard constraints in NNs through differentiable optimization. While there are multiple different approaches to do this (de Avila Belbute-Peres et al., 2018; Donti et al., 2021; Amos & Kolter, 2017), our procedure here is most similar to Négiar et al. (2023). Let $f_\theta : \phi \mapsto \mathbf{b}$, where $\mathbf{b}$ is a set of $N$ scalar-valued functions: $\mathbf{b} = [b^0, b^1, \dots b^N]$ and $b^i : \Omega \to \mathbb{R}$. We refer to $\mathbf{b}$ as a set of *basis* functions. The objective of the hard constraint is to find $\omega \in \mathbb{R}^N$ such that $\mathcal{F}_\phi(\mathbf{b} \cdot \omega^T) = \mathbf{0}$. In other words, $\omega$ is a linear combination of basis functions $\mathbf{b}$ such that the PDE is satisfied. We can solve for $\omega$ using an iterative non-linear least squares solver, such as Levenberg-Marquardt (Nielsen & Madsen, 2010). In practice, the non-linear least squares solver samples a set of $x_1, \dots, x_m$ points in the spatiotemporal domain to evaluate $\mathbf{b}$. This yields $m$ equations of the form $(\mathcal{F}_\phi(\mathbf{b} \cdot \omega^T))(x_i) = \mathbf{0}$. After solving the non-linear least squares problem, the final solution operator is $f_\theta(\phi)(x) \cdot \omega^T$ for $x \in \Omega$, where $f_\theta$ is a NN parameterized by $\theta$.

**Training Physics-Informed Hard Constraints.** Traditional auto-differentiation systems construct a computational graph, where each operation is performed in a "forward pass," and gradients are computed in a "backward pass" in reverse order using the chain rule. Training a NN through backpropagation with the output of an iterative non-linear least squares solvers requires the solver to be differentiable. However, iterative non-linear least squares solvers are not differentiable without unrolling each of the iterations. Unrolling the solver is both computationally expensive (i.e., the computation graph grows) and requires storing all intermediate values in the computation graph.

**Physics-Informed Hard Constraints with Implicit Differentiation.** Implicit differentiation, using the implicit function theorem (Blondel et al., 2022), serves as an alternative to standard auto-differentiation. It performs a second non-linear least squares solve, bypassing the need to unroll the computation graph of the iterative solver. This forgoes the need to store the entire computational history of the solver. Next, we define how we use the implicit function theorem in the context of training a NN via enforcing our physics-informed hard constraints.

The non-linear least squares solver finds $\omega$ such that $\mathcal{F}_\phi(\mathbf{b} \cdot \omega^T) = \mathbf{0}$. $\mathcal{F}_\phi(\mathbf{b} \cdot \omega^T)$ has a non-singular Jacobian $\partial_\mathbf{b} \mathcal{F}_\phi(\mathbf{b} \cdot \omega^T)$. By the implicit function theorem, there exists open subsets $S_\mathbf{b} \subseteq \mathbb{R}^{m \times N}$ and $S_\omega \subseteq \mathbb{R}^N$ containing $\mathbf{b}$ and $\omega$, respectively. There also exists a unique continuous function $z^* : S_\mathbf{b} \to S_\omega$ such that the following properties hold true:

$$\omega = z^*(\mathbf{b}). \qquad \qquad \text{Property (1)}$$

$$\mathcal{F}_\phi(\mathbf{b} \cdot z^*(\mathbf{b})^T) = \mathbf{0}. \qquad \qquad \text{Property (2)}$$

$$z^* \text{ is differentiable on } S_\mathbf{b}. \qquad \qquad \text{Property (3)}$$

Our goal is to find $\frac{\partial \mathcal{F}_\phi(\mathbf{b} \cdot \omega^T)}{\partial \theta}$, where $\theta$ corresponds to NN parameters. This enables us to perform gradient descent on $\theta$ to minimize the loss $||\mathcal{F}_\phi(f_\theta(\phi) \cdot \omega^T)||_2^2$, minimizing the PDE residual. To compute the gradient of the PDE residual, we use Property (1) and differentiate Property (2):

$$\frac{\partial \mathcal{F}_\phi(\mathbf{b} \cdot z^*(\mathbf{b})^T)}{\partial \theta} = \underbrace{\frac{\partial \mathcal{F}_\phi(\mathbf{b} \cdot \omega^T)}{\partial \mathbf{b}} \cdot \frac{\partial \mathbf{b}}{\partial \theta} + \frac{\partial \mathcal{F}_\phi(\mathbf{b} \cdot z^*(\mathbf{b})^T)}{\partial z^*(\mathbf{b})}}_{\text{computed via auto-diff.}} \cdot \frac{\partial z^*(\mathbf{b})}{\partial \theta} = \mathbf{0}. \qquad (1)$$

Eq. 1 defines a system of equations, where $\frac{\partial z^*(\mathbf{b})}{\partial \theta}$ is unknown. We can use the same non-linear least squares solver as the forward pass, concluding the backward pass of training the NN through the hard constraint. Further details on the non-linear least squares solve in Eq. 1 can be found in §A.

**Physics-Informed Hard Constraints with Mixture-of-Experts (MoE).** Until now, we have focused on a single hard constraint with one forward non-linear least squares solve to predict a solution, and one backward solve to train the NN. However, using a single global hard constraint poses multiple challenges. In complicated dynamical systems, the global behavior may drastically vary across the domain, making it difficult for the hard constraint to converge to the right solution. Instead, using multiple constraints over localized locations on the domain can be beneficial to improve constraint adherence on non-sampled points on the mesh. Unfortunately, increased sampling on the mesh is not as simple as using a larger $m$ because the non-linear least squares solve to compute $\omega$ grows with the number of basis functions $N$ and the number of sampled points $m$. As a result, there exists a max $m$ and $N$ after which it is impractical to directly solve the entire system. This can be mitigated by using a smaller batch size, but as we will show, comes at the cost of training stability (Smith et al., 2018; Keskar et al., 2016).

To overcome these challenges, we develop a Mixture-of-Experts (MoE) approach to improve the accuracy and efficiency of physics-informed hard constraints. Suppose we have $K$ experts. The spatiotemporal domain $\Omega$ is partitioned into $K$ subsets $\Omega_k$, for $k = 1 \ldots K$, corresponding to each expert. Each expert individually solves the constrained optimization problem $(\mathcal{F}_\phi(f_\theta(\phi) \cdot \omega_k^T))(x_i) = \mathbf{0}$ for $m$ sampled points $x_i \in \Omega_k$ using a non-linear least squares solver. Because the weighting $\omega_k$ is computed locally, the resulting prediction $\mathbf{b} \cdot \omega_k^T$ is a linear weighting of the global basis functions tailored to the expert's domain $\Omega_k$, which we find leads to more stable training and faster convergence. Each expert is also provided with the global initial and boundary conditions. Given a fixed initial and boundary conditions, the solution that satisfies $\mathcal{F}_\phi(u)$ may be solved point-wise, and solving the constraint over each expert's domain is equivalent to satisfying the constraint globally.

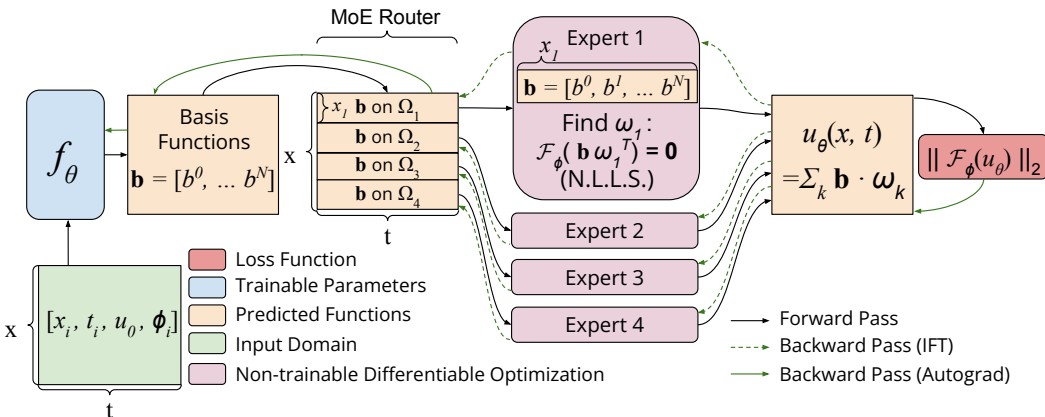

Figure 1: **Schematic of PI-HC-MoE in the 2D case.** PI-HC-MoE is provided with the spatiotemporal grid and any PDE parameters (e.g., initial conditions, viscosity, Reynolds Number). $f_\theta$ is a NN parameterized by $\theta$ (blue box), which outputs a set of $N$ basis functions $\mathbf{b}$ (left-most orange box). $\mathbf{b}$'s domain, the same as the green box, is partitioned into the domains $\Omega_k$ of each expert by the MoE router. Each expert (purple boxes) solves the non-linear least squares problem defined by $\mathcal{F}_\phi(\mathbf{b} \cdot \omega_k^T) = \mathbf{0}$. The resulting $\omega_k$ values are used to produce a final solution $u_\theta = \Sigma_k \mathbf{b} \cdot \omega_k$. Finally, a loss is computed using the $L_2$-norm of the PDE residual (red box). We denote the forward pass with black arrows and the backwards pass with green arrows. Solid green arrows indicate the use of traditional auto-differentiation, while dashed green arrows denote implicit differentiation.

There are multiple potential choices for constructing a domain decomposition $\Omega_k$, and the optimal choice is problem dependent. As a simple 2D example, Fig. 1 uses non-overlapping uniform partitioning along the $x$ dimension. We refer to the center orange boxes, which perform the domain decomposition for the experts, as the MoE router. There are two important nuances to Fig. 1. First, $f_\theta$, the function mapping $\phi$ to the basis functions, is a mapping shared by all experts. Our setup is agnostic to the choice of $f_\theta$, which can be any arbitrary NN. We use Fourier Neural Operator (FNO) (Li et al., 2021a;b), due to its popularity and promising results. Second, each expert performs an iterative non-linear least squares solve to compute $\omega_k$, and the final output is the concatenated outputs across experts. This leads to $k$ independent non-linear least squares solves, which can be parallelized across multiple GPUs.

At test time, the same domain decomposition and router is used. The number of sampled points, $m$, and the parameters of the non-linear least squares solver may be changed (e.g., tolerance), but the domain decomposition remains fixed.

**Forward and backwards pass in Mixture-of-Experts.** In PI-HC-MoE, the forward pass is a domain decomposition in the spatiotemporal grid (see §B for an example forward pass). The domain of each basis function is partitioned according to the MoE router. During the backward pass, each expert needs to perform localized implicit differentiation. We extend this formulation from the single hard constraint to the MoE case. Each expert individually computes $\frac{\partial z^*(\mathbf{b})}{\partial \theta}$ over $\Omega_k$. The MoE router reconstructs the overall Jacobian, given individual Jacobians from each expert. To illustrate this, consider a decomposition along only one axis (e.g., spatial as in Fig. 1). The backward pass must reconstruct the Jacobian across all experts:

$$\partial_\theta z^*(\mathbf{b}) = [\partial_\theta z_1^*(\mathbf{b}) \quad \partial_\theta z_2^*(\mathbf{b}) \quad \ldots \quad \partial_\theta z_K^*(\mathbf{b})]. \tag{2}$$

The realization of the reconstruction operation is performed using an indicator function and summing across Jacobians:

$$\partial_\theta z^*(\mathbf{b}) = \Sigma_{k=1}^K \partial_\theta z_i^*(\mathbf{b}(x)) \cdot \mathbb{1}_{x \in \Omega_k}. \tag{3}$$

Here, $x$ is a point in the domain of expert $k$ (i.e., any spatiotemporal coordinate in $\Omega_k$). Afterwards, the reassembled Jacobian may be used in auto-differentiation as usual.

## 4 RESULTS

We demonstrate our method on two challenging non-linear PDEs: 1D diffusion-sorption (§4.1) and 2D turbulent Navier-Stokes (§4.2). For all problem settings, we use a base FNO architecture.

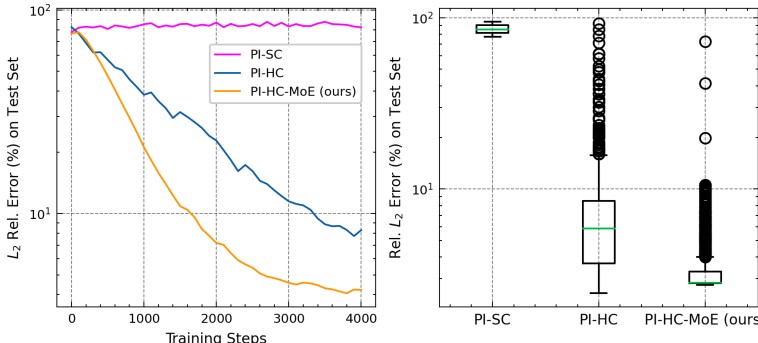

Figure 2: **Relative $L_2$ error on the diffusion-sorption test set.** (Left) The $L_2$ relative error on the test set over training iterations. (Right) The final $L_2$ relative error on the test set using the trained models. PI-HC-MoE converges faster and has greater accuracy than the other settings.

We train this model using our method (Physics-Informed Hard Constraint Mixture-of-Experts, or PI-HC-MoE), and compare to training via a physics-informed soft constraint (PI-SC) and physics-informed hard constraint (PI-HC). For PI-HC and PI-HC-MoE, we use Levenberg-Marquardt as our non-linear least squares solver. For details, see §C.1 and §C.2.

**Data-constrained setting.** In both problems, we exclusively look at a data-constrained setting where we assume that at training time, we have no numerical solver solution data available (i.e., no solution data on the interior of the domain). We do so to mimic real-world settings, where it can be expensive to generate datasets for new problem settings (see §C for further discussion).

**Evaluation details.** The training and test sets contain initial conditions drawn from the same distribution, but the test set initial conditions are distinct from the training set. To evaluate all models, we compute numerical solver solutions to compare relative $L_2$ errors on ML predictions using the test set initial conditions. The PDE residual over training on the validation set for diffusion-sorption and Navier-Stokes is included in §F.For measuring scalability, we use the best trained model for both PI-HC-MoE and PI-HC to avoid ill-conditioning during the non-linear least squares solve. The training step includes the cost to backpropagate and update the model weights. All measurements are taken across 64 training or inference steps on NVIDIA A100 GPUs, and each step includes one batch. We report average per-batch speedup of PI-HC-MoE compared to PI-HC.

### 4.1 1D DIFFUSION-SORPTION

We study the 1D non-linear diffusion-sorption equation. The diffusion-sorption system describes absorption, adsorption, and diffusion of a liquid through a solid. The governing PDE is defined as:

$$\frac{\partial u(x,t)}{\partial t} = \frac{D}{R(u(x,t))} \cdot \frac{\partial^2 u(x,t)}{\partial x^2}, \qquad x \in (0,1), t \in (0,500],$$

$$R(u(x,t)) = 1 + \frac{1-\phi}{\phi} \cdot \rho_s \cdot k \cdot n_f \cdot u(x,t)^{n_f - 1},$$

$$u(0,t) = 1, \quad u(1,t) = D \cdot \frac{\partial u(1,t)}{\partial x}, \qquad \text{(Boundary Conditions)}$$

where $\phi, \rho_s, D, k$ and $n_f$ are constants defining physical quantities. We use the same physical constants as PDEBench (Takamoto et al., 2022) (see §C.1 for further details). For both classical numerical methods and ML methods, the singularity at $u = 0$ poses a hard challenge. The solution trajectory and differential operator are highly non-linear. A standard finite volume solver requires approximately 60 seconds to compute a solution for a 1024×101 grid (Takamoto et al., 2022).

**Problem Setup.** We use initial conditions from PDEBench. Each solution instance is a scalar-field over 1024 spatial and 101 temporal points, where $T = 500$ seconds (see §C.1 for further details).

For PI-HC-MoE, we use $K = 4$ experts and do a spatial decomposition with $N = 16$ basis functions (see §B for details on inference). Each expert enforces the constraint on a domain of 256 (spatial) × 101 (temporal). Over the $256 \times 101$ domain, each expert samples 20k points, leading to a total of 80k points where the PDE constraint is enforced on the domain. PI-HC uses an identical model with

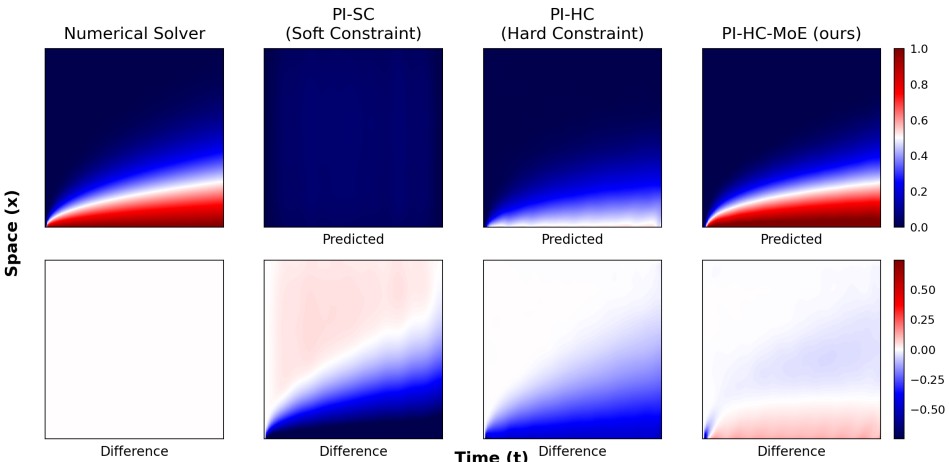

Figure 3: **Predicted solution for the diffusion-sorption equation.** (Top) Visualizations of the numerical solver solution and ML predictions for the soft constraint (PI-SC), hard constraint (PI-HC), and PI-HC-MoE. (Bottom) Difference plots of the ML predicted solutions compared to the numerical solver solution. White denotes zero. PI-SC is unable to recover the dynamics or scale of the solution. PI-HC is able to recover some information, but fails to capture the full dynamics. PI-HC-MoE is able to recover almost all of the solution and has the lowest error.

20k sampled points in the hard constraint. This represents the maximum number of points that we can sample with PI-HC without running out of memory, with a stable batch size. In order to sample more points, a reduction in batch size is required. Our final batch size for PI-HC is 6, and we find that any reduction produces significant training instability, with a higher runtime (see Fig. 4). Thus, our baseline PI-HC model uses the maximum number of sampled points that gives the best training stability. All models are trained with a fixed computational budget of 4000 training iterations.

**Results.** We summarize our results in Fig. 2 and Fig. 3. In Fig. 2 (left), we plot the $L_2$ relative percent error on the test set over training steps for PI-SC, PI-HC, and PI-HC-MoE. On the right, we plot the trained model $L_2$ relative percent error on the test set of the final trained models. PI-SC is unable to converge to a reasonable solution, reflected in the high $L_2$ relative percent error ($\mathbf{85.93\%} \pm \mathbf{5.07\%}$). While PI-HC is able to achieve lower error than PI-SC at $\mathbf{7.55\%} \pm \mathbf{8.10\%}$, it does worse than PI-HC-MoE ($\mathbf{3.60\%} \pm \mathbf{2.93\%}$). In Fig. 3, we show the predicted ML solutions for PI-SC, PI-HC, and PI-HC-MoE, and the difference between the predicted solution and the numerical solver solution (white indicates zero difference). PI-SC is unable to converge to a reasonable solution, and PI-HC, while closer, is unable to capture the proper dynamics of diffusion-sorption. PI-HC-MoE has the closest correspondence to the numerical solution. Additionally, we explore PI-HC-MoE 's generalization to unseen timesteps (§D) and assess the quality of basis functions (§E).

**Scalability.** We compare the scalability of PI-HC-MoE to PI-HC. For evaluation, we plot the number of sampled points on the interior of the domain against the execution time during training and inference, in Fig. 4. PI-HC-MoE maintains near constant execution time as the number of points increases, while PI-HC becomes significantly slower as the number of sampled points crosses 50k

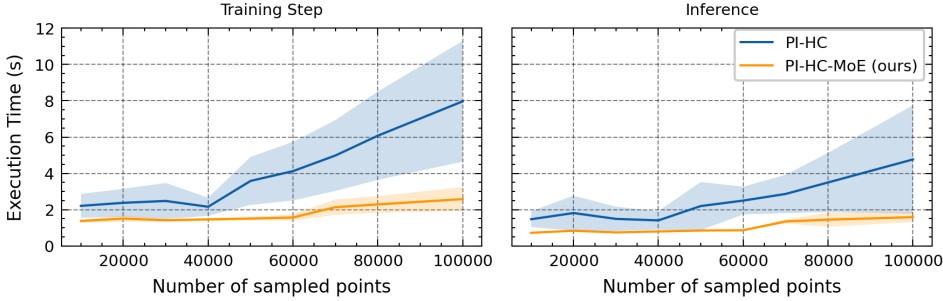

Figure 4: **Runtime of PI-HC and PI-HC-MoE on diffusion-sorption.** The time to perform a single training (left) and inference (right) step as the number of constrained sampled points increases.

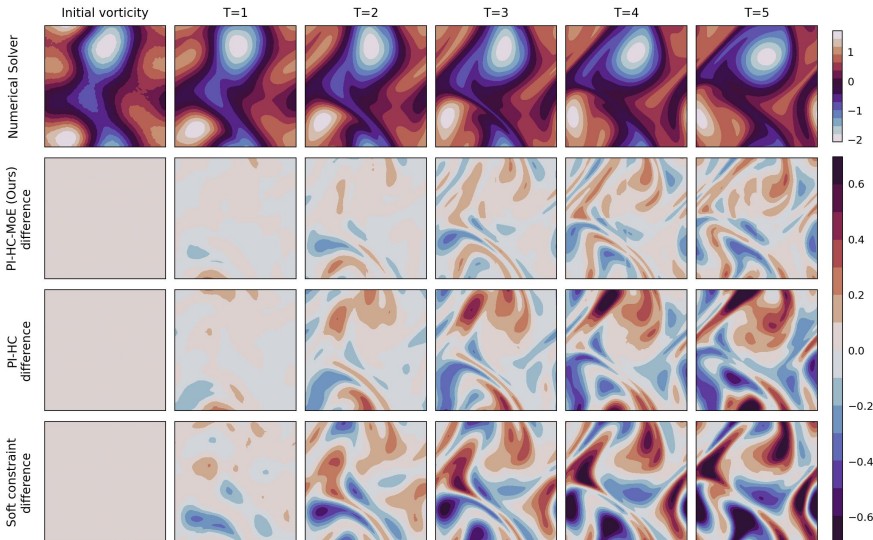

Figure 5: **Predicted solution for 2D Navier-Stokes.** From top to bottom: (Row 1) Initial vorticity and its evolution as $T$ increases, computed via a numerical solver. The errors of PI-HC-MoE (Row 2), PI-HC (Row 3), and PI-SC (Row 4) are visualized for corresponding $T$, where the difference in the predicted solution is shown with respect to the numerical solver. Darker colors indicate higher error. Both PI-SC and PI-HC exhibit greater error compared to PI-HC-MoE, especially at later $T$.

points. PI-HC-MoE provides a training speedup of $\mathbf{1.613}\times$ ($10^2$ sampled points) to $\mathbf{3.189}\times$ ($10^5$ sampled points). Notably, PI-HC has a higher standard deviation as the number of sampled points increases. Because PI-HC-MoE partitions the number of sampled points, the individual constraint solved by each expert converges faster. As a result, PI-HC-MoE is more consistent across different data batches and has a much lower variation in runtime.

Note that a standard finite volume solver takes about 60 seconds to solve for a solution (Takamoto et al., 2022), and so PI-HC and PI-HC-MoE both have inference times faster than a numerical solver. However, PI-HC-MoE is $\mathbf{2.030}\times$ ($10^2$ sampled points) to $\mathbf{3.048}\times$ ($10^5$ sampled points) faster at inference than PI-HC, while also having significantly lower error.

### 4.2 2D NAVIER-STOKES

The Navier-Stokes equations describe the evolution of a fluid with a given viscosity. We study the vorticity form of the 2D periodic Navier-Stokes equation, where the learning objective is to learn the scalar field vorticity $w$.

$$\frac{\partial w(t,x,y)}{\partial t} + u(t,x,y) \cdot \nabla w(t,x,y) = \nu \Delta w(t,x,y), \quad t \in [0,T], \quad (x,y) \in (0,1)^2 \quad (4)$$

$$w = \nabla \times u, \quad \nabla \cdot u = 0,$$

$$w(0,x,y) = w_0(x,y), \quad \text{(Boundary Conditions)}$$

where $u$ is the velocity vector field, and $\nu$ is the viscosity. Similar to Li et al. (2021b) and Raissi et al. (2019), we study the vorticity form of Navier-Stokes to reduce the problem to a scalar field prediction, instead of predicting the vector field $u$. In our setting, we use a Reynolds number of $1e^4$ ($\nu = 1e-4$), representing turbulent flow. Turbulent flow is an interesting and challenging problem due to the complicated evolution of the fluid, which undergoes irregular fluctuations. Many engineering and scientific problems are interested in the turbulent flow case (e.g., rapid currents, wind tunnels, and weather simulations (Nieuwstadt et al., 2016)).

**Problem setup.** The training set has a resolution of 64 ($x$) × 64 ($y$) × 64 ($t$). Both the training and test sets have a trajectory length of $T = 5$ seconds. The test set has a resolution of 256 ($x$) × 256 ($y$) × 64 ($t$). For both PI-HC and PI-HC-MoE, we use 64 basis functions. We use $K = 4$ experts and perform a 2 ($x$) ×2 ($y$) ×1 ($t$) spatiotemporal decomposition. Each expert receives the full temporal grid with $\frac{1}{4}$ of the spatial grid (i.e., $32 \times 32 \times 64$ input), and samples 20k points during the constraint step. For the full data generation parameters and architecture details, see §C.2.

**Results.** We visualize representative examples from the $64^3$ test set in Fig. 5, comparing PI-SC, PI-HC, and PI-HC-MoE . In Fig. 5, grey represents zero error. PI-SC is the worst performing model, with significant errors appearing by $T = 2$. While PI-HC captures most of the dynamics at earlier time steps ($T = 1$, $T = 2$), especially compared to PI-SC, PI-HC struggles to adequately capture the behavior of later time steps. In particular, PI-HC fails to capture the evolution of fine features at $T = 5$. On the $256 \times 256 \times 64$ test set, PI-SC attains a relative $L_2$ error of $\mathbf{18.081\%} \pm \mathbf{3.740\%}$. PI-HC ($\mathbf{11.754\%} \pm \mathbf{2.951\%}$) and PI-HC-MoE ($\mathbf{8.298\%} \pm \mathbf{2.345\%}$) both achieve a lower relative $L_2$ error. PI-HC-MoE consistently attains the lowest error across all three resolutions and has lower variance in prediction quality (box plots visualized in appendix §C.3).

**Scalability.** Similar to the diffusion-sorption case, we evaluate the scalability of our approach. We compare the scalability of PI-HC-MoE to standard differentiable optimization (i.e., PI-HC) for both the training and test steps across a different number of sampled points in Fig. 6. The training and inference steps are performed with a batch size of 2. At training time, PI-HC scales quadratically with respect to the number of sampled points, and has high variance in per batch training and inference time. PI-HC has a harder constraint system to solve, reflected in the increase in training

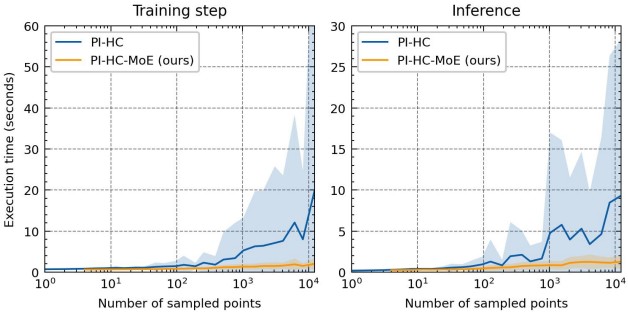

Figure 6: **Runtime of PI-HC and PI-HC-MoE on 2D Navier-Stokes.** We plot the time for a single training (left) and inference (right) iteration. PI-HC-MoE is significantly faster for both, and scales sublinearly. PI-HC has much higher execution times and a large standard deviation.

time. Across both training and inference steps, PI-HC-MoE scales sublinearly and, in practice, remains constant with respect to the number of sampled points. For training, PI-HC-MoE provides a $\mathbf{2.117\times}$ ($10^2$ sampled points) to $\mathbf{12.838\times}$ ($10^4$ sampled points) speedup over PI-HC. At inference time we see speedups of $\mathbf{2.538\times}$ ($10^2$ sampled points) to $\mathbf{12.864\times}$ ($10^4$ sampled points).

### 4.3 FINAL TAKEAWAYS

PI-HC-MoE has lower $L_2$ relative error on the diffusion-sorption and Navier-Stokes equations, when compared to both PI-SC and PI-HC. Enabled by the MoE setup, PI-HC-MoE is better able to capture the features for both diffusion-sorption and Navier-Stokes. PI-HC-MoE is also more scalable than the standard differentiable optimization setting represented by PI-HC. Each expert locally computes $\omega_k$, allowing for greater flexibility when weighting the basis functions. The experts are better able to use the local dynamics, while satisfying the PDE globally. PI-HC is limited to a linear combination of the basis functions, whereas PI-HC-MoE is able to express piece-wise linear combinations.

Another reason we find PI-HC-MoE to outperform PI-HC is through the stability of training. This manifests in two different ways. First, because PI-HC-MoE is more scalable, we are able to use a larger batch size, which stabilizes the individual gradient steps taken. Second, for a given total number of sampled points, the non-linear least square solves performed by PI-HC-MoE are smaller than the global non-linear least squares performed by the hard constraint. Specifically, the size of the constraint for $K$ experts is $\frac{1}{K}$ that of PI-HC. This results in an easier optimization problem, and PI-HC-MoE converges quicker and with greater accuracy.

## 5 CONCLUSION

We present the physics-informed hard constraint mixture-of-experts (PI-HC-MoE) framework, a new approach to scale hard constraints corresponding to physical laws through an embedded differentiable optimization layer. Our approach deconstructs a differentiable physics hard constraint into smaller experts, which leads to better convergence and faster run times. On two challenging, highly non-linear systems, 1D diffusion-sorption and 2D Navier-Stokes equations, PI-HC-MoE achieves significantly lower errors than standard differentiable optimization using a single hard constraint, as well as soft constraint penalty methods.

**Acknowledgements.** This work was initially supported by Laboratory Directed Research and Development (LDRD) funding under Contract Number DE-AC02-05CH11231. It was then supported by the U.S. Department of Energy, Office of Science, Office of Advanced Scientific Computing Research, Scientific Discovery through Advanced Computing (SciDAC) program under contract No. DE-AC02-05CH11231, and in part by the Office of Naval Research (ONR) under grant N00014-23-1-2587. This research used resources of the National Energy Research Scientific Computing Center (NERSC), a U.S. Department of Energy Office of Science User Facility located at Lawrence Berkeley National Laboratory, operated under Contract No. DE-AC02-05CH11231. We also thank Geoffrey Négiar, Sanjeev Raja, and Rasmus Malik Høegh Lindrup for their valuable feedback and discussions.

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

## A  IFT: BACKWARD PASS

We expand on the training procedure described in §3, and additional details on training the model using the implicit function theorem.

Recall that for basis functions $\mathbf{b} = [b^0, b^1, \ldots b^N]$ and $b^i : \Omega \to \mathbb{R}$, the non-linear least squares solver finds $\omega \in \mathbb{R}^N$ such that $\mathcal{F}_\phi(\mathbf{b} \cdot \omega^T) = \mathbf{0}$. If the Jacobian $\partial_\mathbf{b} \mathcal{F}_\phi(\mathbf{b} \cdot \omega^T)$ is non-singular, then the implicit function theorem holds. There exists open sets $S_\mathbf{b} \subseteq \mathbb{R}^{m \times N}$, $S_\omega \subseteq \mathbb{R}^N$, and function $z^* : S_\mathbf{b} \to S_\omega$. $S_\mathbf{b}$ and $S_\omega$ contain $\mathbf{b}$ and $\omega$. $z^*$ has the following properties:

$$\omega = z^*(\mathbf{b}). \qquad \qquad \text{Property (1)}$$

$$\mathcal{F}_\phi(\mathbf{b} \cdot z^*(\mathbf{b})^T) = \mathbf{0}. \qquad \qquad \text{Property (2)}$$

$$z^* \text{ is differentiable on } S_\mathbf{b}. \qquad \qquad \text{Property (3)}$$

During gradient descent, by the chain rule, we need to know $\frac{\partial z^*(\mathbf{b})}{\partial \theta}$, which is not readily available by auto-differentiation. By differentiating property (2), we get Eq. 1:

$$\frac{\partial \mathcal{F}_\phi(\mathbf{b} \cdot z^*(\mathbf{b})^T)}{\partial \theta} = \frac{\partial \mathcal{F}_\phi(\mathbf{b} \cdot \omega^T)}{\partial \mathbf{b}} \cdot \frac{\partial \mathbf{b}}{\partial \theta} + \frac{\partial \mathcal{F}_\phi(\mathbf{b} \cdot z^*(\mathbf{b})^T)}{\partial z^*(\mathbf{b})} \cdot \frac{\partial z^*(\mathbf{b})}{\partial \theta} = \mathbf{0}. \qquad (1)$$

Rearranging Eq. 1, we can produce a new system of equations.

$$\frac{\partial \mathcal{F}_\phi(\mathbf{b} \cdot z^*(\mathbf{b})^T)}{\partial z^*(\mathbf{b})} \cdot \frac{\partial z^*(\mathbf{b})}{\partial \theta} = -\frac{\partial \mathcal{F}_\phi(\mathbf{b} \cdot \omega^T)}{\partial \mathbf{b}} \cdot \frac{\partial \mathbf{b}}{\partial \theta}$$

$$\frac{\partial z^*(\mathbf{b})}{\partial \theta} = \left[ \frac{\partial \mathcal{F}_\phi(\mathbf{b} \cdot z^*(\mathbf{b})^T)}{\partial z^*(\mathbf{b})} \right]^{-1} \cdot -\frac{\partial \mathcal{F}_\phi(\mathbf{b} \cdot \omega^T)}{\partial \mathbf{b}} \cdot \frac{\partial \mathbf{b}}{\partial \theta} \qquad (5)$$

The unknown that we need to solve for now is the desired quantity, $\frac{\partial z^*(\mathbf{b})}{\partial \theta}$. The expressions on the right-hand side, $\frac{\partial \mathcal{F}_\phi(\mathbf{b} \cdot \omega^T)}{\partial \mathbf{b}} \cdot \frac{\partial \mathbf{b}}{\partial \theta}$ and $\frac{\partial \mathcal{F}_\phi(\mathbf{b} \cdot z^*(\mathbf{b})^T)}{\partial z^*(\mathbf{b})}$, can be computed via auto-differentiation. This then gives us a system of equations which involves a matrix inverse. If matrix inversion was computationally tractable, we could explicitly solve for $\frac{\partial z^*(\mathbf{b})}{\partial \theta}$. Instead, the system of equations is solved using the same non-linear least squares solver as in the forward pass, allowing us to approximate the matrix inverse, and yields $\frac{\partial z^*(\mathbf{b})}{\partial \theta}$. Now, we can compute $\frac{\partial \mathcal{F}_\phi(\mathbf{b} \cdot \omega^T)}{\partial \theta}$ and train the full model end-to-end.

## B  EXAMPLE INFERENCE PROCEDURE

To demonstrate the inference procedure of PI-HC-MoE, we walk through the forward pass of the 1D diffusion-sorption problem setting, starting from PI-HC and generalizing to the PI-HC-MoE case.

**Model Input and Output.**  In both PI-HC and PI-HC-MoE, the underlying FNO model is provided with both the initial condition and underlying spatiotemporal grid. Each initial condition is a 1024 dimensional vector (the chosen discretization), representing the spatial coordinates, and is broadcasted across time (t = 500 s), leading to an input of $1024 \times 101$ for the initial condition. The broadcasted initial condition is concatenated with the spatiotemporal grid with shape $1024 \times 101 \times 2$, for a final input spanning the entire grid with 3 channels ($1024 \times 101 \times 3$). The inference procedure of the underlying base NN achitecture is unchanged. In our case, the base NN architecture produces a $1024 \times 101 \times N$ tensor representing $N$ basis functions evaluated over the spatiotemporal domain.

**PI-HC (Single Constraint).**  Each constraint samples $M$ points and solves the PDE equations. Additionally, the input initial condition (1024) and initial condition of the basis functions ($1024 \times N$)

are provided to the solver. Similarly, the boundary conditions are provided and diffusion-sorption has two boundary conditions at $x = 0$ and $x = 1$, which leads to a boundary condition tensor of $2 \times 101 \times N$. The optimal $\omega$ $(N)$ computed by the non-linear least squares solver satisfies the PDE equations on the $M$ sampled points, as well as the initial and boundary conditions. Finally, $\omega$ is used to compute the entire scalar field by performing a matrix multiply between the basis functions and $\omega$ ($1024 \times 101 \times N \cdot N = 1024 \times 101$ predicted scalar field).

**PI-HC-MoE (Multiple Constraints).** The MoE router performs a domain decomposition. Assuming $K = 4$ spatial experts, the $1024 \times 101 \times N$ basis function evaluation is deconstructed into four $256 \times 101 \times N$ tensors. Each expert performs a similar computation as PI-HC, and notably, the global initial ($1024 \times N$) and boundary ($2 \times 101 \times N$) conditions are provided to each expert's non-linear least squares solve. After computing a localized weighting $\omega_k$, each expert returns a $256 \times 101$ scalar field representing the local predicted solution field. Finally, the MoE router reverses the domain decomposition, assembling the four $256 \times 101$ matrices into a final $1024 \times 101$ prediction. This final prediction represents the complete constrained output.

## C  PROBLEM SETTING DETAILS

We use Jax (Bradbury et al., 2018) with Equinox (Kidger & Garcia, 2021) to implement our models. For diffusion-sorption, we use Optimisix's (Rader & Kidger) Levenberg-Marquardt solver and for Navier-Stokes, we use JaxOpt's (Blondel et al., 2022) Levenberg-Marquardt solver. In both the PI-HC and PI-HC-MoE cases, we limit the number of iterations to 50.

**Data-constrained setting.** We exclusively look at the data-starved regime. As a motivating example, in the 1D diffusion-sorption, PDEBench (Takamoto et al., 2022) provides a dataset of 10k solution trajectories. For a difficult problem like diffusion-sorption, the numerical solver used in PDEBench takes over one minute to solve for a solution trajectory. A dataset of 10k would require almost a week of sequential compute time. Additionally, PDEBench only provides a dataset using one set of physical constants (e.g., porosity). Different diffusion mediums require different physical constants, making the problem of generating a comprehensive dataset very computationally expensive. For these reasons, we focus on only training the NN via the PDE residual loss function.

### C.1  ADDITIONAL DETAILS: DIFFUSION-SORPTION

**Physical Constants and Dataset Details.** $\phi = 0.29$ is the porosity of the diffusion medium. $\rho_s = 2880$ is the bulk density. $n_f = 0.874$ is Freundlich's exponent. Finally, $D = 5 \cdot 10^{-4}$ is the effective diffusion coefficient.

The training set has 8000 unique initial conditions and the test set has 1000 initial conditions, distinctly separate from the training set. We use the same initial conditions from PDEBench (Takamoto et al., 2022).

**Architecture and Training Details.** The base model we use is an FNO (Li et al., 2021a) architecture with 5 Fourier layers, each with 8 modes and 64 hidden feature representation. We use a learning rate of $1e^{-3}$ with an exponential decay over 4000 training iterations. The tolerance of the Levenberg-Marquardt is set to $1e^{-4}$.

### C.2  ADDITIONAL DETAILS: 2D NAVIER-STOKES

**Data generation.** All initial conditions for the training and test set are are generated from a 2D Gaussian random field with a periodic kernel and a length scale of 0.8. The training set has 4000 initial conditions with resolution of 64 $(x) \times 64$ $(y) \times 64$ $(t)$. Both the training and test sets have a trajectory length of $T = 5$ seconds. We generate a test set of 100 solutions with a resolution of $256 \times 256$ (spatial) and 64 time steps. The error is measured on the original resolution solutions, as well as spatially subsampled versions ($128 \times 128$, $64 \times 64$).

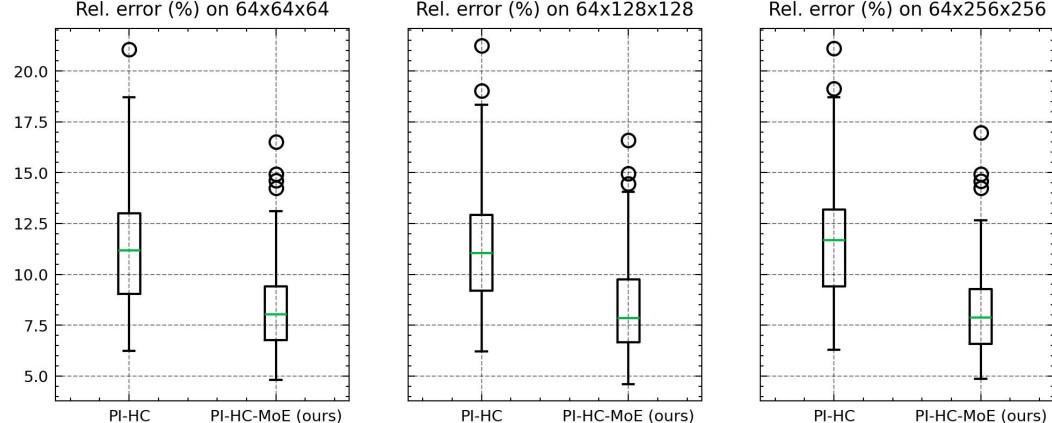

Figure 7: **Final $L_2$ relative error % on 3 test sets for 2D Navier-Stokes.** Both PI-HC and PI-HC-MoE train on a $64 \times 64 \times 64$ spatiotemporal grid. PI-HC-MoE generalizes to spatial resolutions better than PI-HC with lower variance.

**Architecture details.** We use an FNO with 8 Fourier layers and 8 modes as the base NN architecture. The Levenberg-Marquardt solver tolerance is set to $1e^{-7}$ and we use a learning rate of $1e^{-3}$ with an exponential decay over 20 epochs and a final learning rate of $1e^{-4}$.

### C.3 ADDITIONAL RESULTS: 2D NAVIER-STOKES

In Fig. 7, we include a box plot showing the test set errors on three different spatial resolutions ($256^2$, $128^2$, $64^2$). The $64^2$ and $128^2$ solutions are subsampled from the $256^2$ solution. PI-HC-MoE outperforms PI-HC on all test sets.

### C.4 ADDITIONAL DETAILS: SCALABILITY RESULTS

When conducting our scalability analysis, we measure the per-batch time across 64 training and test steps. We observe large standard deviations in our speedup numbers and provide a short discussion here. In Fig. 4 and Fig. 6, we plot our measured runtimes as a function of the number interior points constrained. In both the diffusion-sorption and Navier-Stokes settings, we observe that PI-HC has large standard deviations at a fixed number of sampled points. We attribute this variation to the difficulty of the constraint performed by PI-HC, which causes high fluctuations in the number of non-linear least squares iterations performed. As a result, when we calculate speedup values, the speedup values also have high std deviations. In the main text, we report the mean speedup across all batches for a fixed number of sampled points; for completeness we include the standard deviations here.

In the 1D diffusion-sorption setting, we see standard deviations at inference from 0.587 ($10^2$ sampled points) to 1.890 ($10^4$ sampled points). For training, the standard deviations vary from 0.481 ($10^2$ sampled points) to 1.388 ($10^4$ sampled points). Analogously, in 2D Navier-Stokes, we see standard deviations from 3.478 ($10^2$ sampled points) to 35.917 ($10^4$ sampled points) for inference. For training, we measure standard deviations from 2.571 ($10^2$ sampled points) to 47.948 ($10^4$ sampled points). The large standard deviations are a reflection of the standard deviations of the runtime numbers for PI-HC for both problems.

## D ADDITIONAL EXPERIMENT: TEMPORAL GENERALIZATION

We explore PI-HC-MoE's generalization to temporal values not in the training set, for the diffusion-sorption case.

**Problem Setup.** In our original problem setting, we predict diffusion-sorption from $t = 0$ seconds to $t = 500$ seconds. Here, in order to test generalization to unseen temporal values, we truncate

the training spatiotemporal grids to $t < 400$ (s), i.e., we only train on $t \in [0, 400]$. After training, PI-SC, PI-HC, and PI-HC-MoE all predict the solution up to $t = 500$ seconds. We compute the $L_2$ relative error error in prediction for $t > 400$ (s) (the domain that was unseen during training). The parameters for all 3 models are identical to §4.1. PI-HC-MoE uses $K = 4$ spatial experts.

**Results.** Fig. 8 shows the $L_2$ relative error for $t > 400$ (s) on the validation set. Both PI-HC ($\mathbf{48.047\% \pm 1.23\%}$) and PI-HC-MoE ($\mathbf{14.79\% \pm 3.00\%}$) outperform PI-SC ($\mathbf{805.58\% \pm 19.06\%}$). While PI-HC is able to do better than PI-SC, PI-HC-MoE is able to generalize the best to future time steps, even without any training instances.

We hypothesize that one reason for the performance improvement of PI-HC and PI-HC-MoE comes from test-time constraint enforcement, demonstrating the benefit of hard constraints at inference time. Similar to the results in §4.1, PI-HC-MoE is able better generalize much better to the unseen temporal domain than PI-HC, while also being much more efficient at both training and inference time.

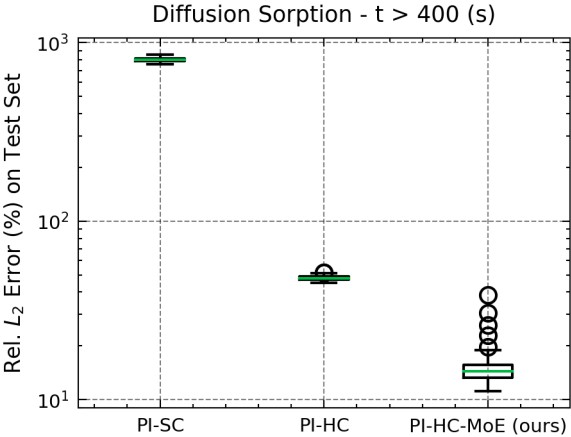

Figure 8: **Relative $L_2$ error on unseen temporal values ($t > 400$ (s)) for diffusion-sorption.** All models are trained on $t \in [0, 400]$, and then predict the solution up to $t = 500$ seconds. The relative $L_2$ error is plotted for $t > 400$ seconds. PI-SC is unable to generalize to out-of-distribution temporal values. PI-HC is able to do better, but still has high error, while PI-HC-MoE has the lowest error.

# E  ABLATION: EVALUATING THE QUALITY OF THE LEARNED BASIS FUNCTIONS

In order to explore the usefulness of our learned basis functions, we conduct an ablation study comparing the result of PI-HC-MoE to a standard cubic interpolation.

**Problem setup.** We use the same PI-HC-MoE model from the diffusion-sorption experiments in §4.1, where each expert constrains 20k sampled points ($K = 4$ experts, 80k total sampled points). The 20k constrained points represents a candidate solution to the PDE where, by the non-linear least squares solve, the PDE equations must be satisfied. Using the 20k points, we perform an interpolation using SciPy's (Virtanen et al., 2020) 2D Clough-Tocher piecewise cubic interpolation. The interpolated solution is assembled in an identical manner to PI-HC-MoE, representing $K = 4$ interpolations, and corresponds to the number of experts.

**Results.** Fig. 9 compares the relative $L_2$ error on the test set between PI-HC-MoE 's learned basis functions and the interpolated solution. PI-HC-MoE outperforms the interpolation scheme, showing that the learned basis functions provide an advantage over interpolation, and encode additional unique information. We also note that it is likely that with fewer sampled points, the performance gap between PI-HC-MoE and interpolation is likely to be higher.

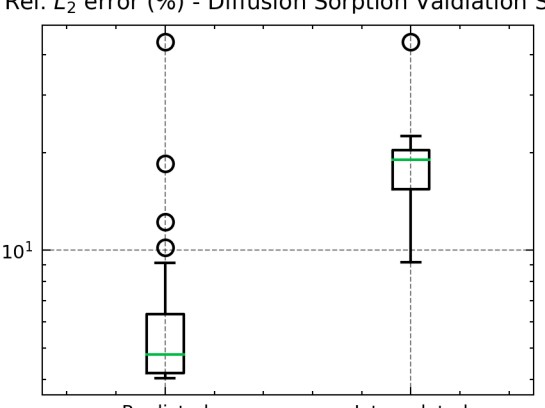

Figure 9: **Relative $L_2$ error on 1D diffusion-sorption's test set for PI-HC-MoE and an interpolated solution.** The learned basis functions by PI-HC-MoE represent a more accurate solution compared to interpolating the constrained points. The error plot shows that even for unconstrained points, PI-HC-MoE is a closer approximation to the numerical solver solution.

## F    PDE RESIDUALS FOR DIFFUSION-SORPTION AND NAVIER-STOKES

We add additional PDE residual loss plots that correspond to the results in §4.1 and §4.2. During the training of all three models (PI-SC, PI-HC, and PI-HC-MoE ), we track the mean PDE residual value across the validation set, plotted in Fig. 10. Across both problem instances, PI-HC-MoE attains the lowest PDE residual, and also has the lowest $L_2$ relative error (see  §4.1). For 1D diffusion-sorption, PI-SC initially starts with low PDE residual before increasing near the end of training. The reason for this is that at the beginning of training, when PI-SC is first initialized, the predicted solution is close to the zero function and trivially satisfies the PDE equations, i.e., both $\frac{\partial u}{\partial t}$ and $\frac{\partial^2 u}{\partial x^2}$ are zero. However, the overall loss function includes adherence to the initial and boundary conditions and PI-SC is unable to find a parameterization that satisfies the initial condition, boundary conditions, and PDE equations.

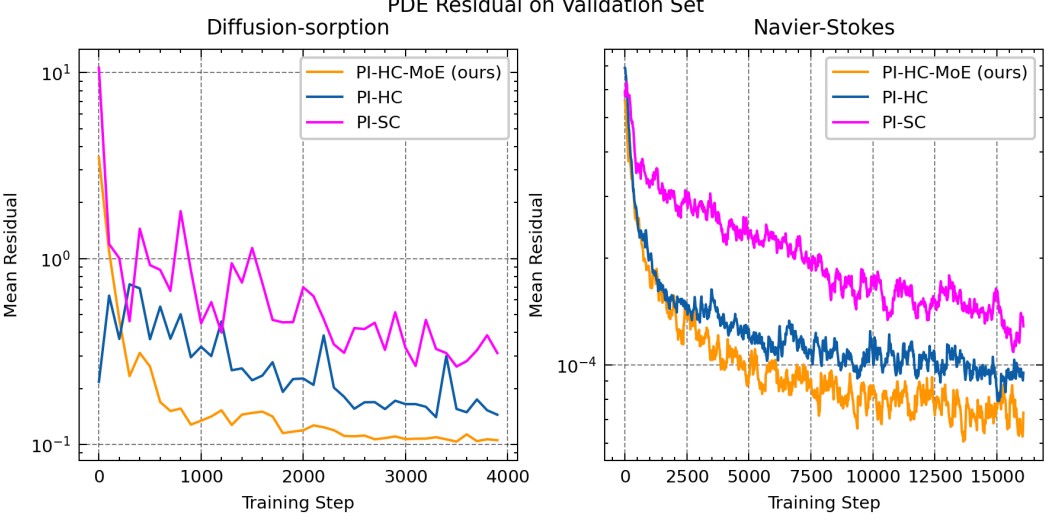

Figure 10: **PDE residual on validation set during training.** The mean PDE residual over the validation set over training for both 1D diffusion-sorption (left) and 2D Navier-Stokes (right). PI-HC-MoE has the lowest PDE residual.

## G    LIMITATIONS AND FUTURE WORK

PI-HC-MoE, though a promising framework for scaling hard constraints, has some limitations.

**Hyperparameters.**    Both PI-HC-MoE and PI-HC are sensitive to hyperparameters. It is not always clear what the correct number of basis functions, sampled points, number of experts, and expert distribution is best suited for a given problem. However, our results show that with only minor hyperparameter tweaking, we were able to attain low error on two challenging dynamical systems. As with many ML methods, hyperparameter optimization is a non-trivial task, and one future direction could be better ways to find the optimal parameters.

**Choice of domain decomposition.**    Currently, PI-HC-MoE performs a spatiotemporal domain decomposition to assign points to experts. A possible future direction is trying new domain decompositions, and methods for allocating experts. It may be the case that different ways of creating expert domains are better suited for different problem settings.

**Base NN architecture.**    In this work, we use FNO as our base architecture for PI-HC-MoE and in the future, it may prove fruitful to try new architectures. To some extent, PI-HC-MoE is limited by the expressivity of the underlying NN architecture which learns and predicts the basis functions. PI-HC-MoE may perform better on certain tasks that we have not yet explored (e.g., super-resolution, auto-regressive training), and future work could explore the application of PI-HC-MoE to new kinds of tasks.

