# OpenReview forum: "Scaling physics-informed hard constraints with mixture-of-experts"
_ICLR.cc/2024/Conference — ICLR 2024 poster_

### Official Review · Reviewer_v3ig · 2023-10-29

**Soundness:** 3 good
**Presentation:** 3 good
**Contribution:** 3 good
**Rating:** 6
**Confidence:** 3

**Summary:**

This work aims to reduce the cost of imposing hard physical constraints in deep neural network training by utilizing Mixture-of-Experts. The proposed approach is scalable and allows multi-GPU training. The improved stability and efficiency of the proposed method over existing approaches are well demonstrated on non-linear systems.

**Strengths:**

(1) The paper is overall well written.

(2) The problem addressed in this paper, which is how to impose physical constraints into large-scale deep neural network training, is of importance.

(3) The improvement of the proposed method over baselines is significant.

**Weaknesses:**

Major: As claimed by the authors in the last paragraph in page 4, the utilizing of Mixture-of-Experts is motivated by two reasons/challenges. (A) In complicated systems, compared with global physical constraints, local constraints are more manageable. (B) The scalability of the non-linear least squares solver impeded the increase in the number of sample points (m), thereby affecting the accuracy of predictions.

Reason A aligns with the phenomenon we have observed in language tasks [1]: data from different domains can interfere with each other and hinder training. However, the authors did not provide evidence for the occurrence of this phenomenon in dynamic systems, nor did they demonstrate that the improvement of the PI-HC-MoE over PI-HC is due to avoiding such interference. One possible experiment to demonstrate the latter is to randomly assign data points to experts and compare it to the allocation based on spatio-temporal coordinates.

Challenge B can be overcome by utilizing MoE. However, a more straightforward solution will be employing scalable non-linear least squares solvers, e.g. [2][3]. The authors didn’t comment on this line of works. If there are suitable solvers available, I would like to see a comparison between PI-HC and PI-HC-MoE with the same number of data points.

Furthermore, the performance of the baseline appears to be inconsistent with the results reported in [4], e.g., 1D non-linear diffusion-sorption equation results, see [4], Appendix E.

Minor: There are several errors and confusions in the use of mathematical symbols in the text, including but not limited to
(1) in section 3, first paragraph, line 1, $0$ should be in bold.
(2) in section 3, first paragraph, line 4, the definition of $\mathcal{G}$ and in the second paragraph, line 4, the definition of $f_{\theta}$, other notation should be used to denote the spaces of $\phi$, $u_\theta$ and $\mathbf{b}$.
(3) In section 3, second paragraph, line 5, $f^i: \phi \rightarrow \mathbb{R}$ should be $f^i: \Omega \rightarrow \mathbb{R}$.
(4) In section 3, second paragraph, third to last line, $f_\theta(\phi(x))$ should be $f_\theta(\phi)(x)$.
(5) In page 4, two lines above Property (1), the definition of $S_{\mathbf{b}}$ and $S_\omega$.

Similar errors are repeated throughout the paper and should be corrected.

[1] Outrageously Large Neural Networks: The Sparsely-Gated Mixture-of-Experts Layer
[2] Scalable Subspace Methods for Derivative-Free Nonlinear Least-Squares Optimization
[3] DeepLM: Large-scale Nonlinear Least Squares on Deep Learning Frameworks using Stochastic Domain Decomposition
[4] PDEBENCH: An Extensive Benchmark for Scientific Machine Learning

**Questions:**

see Weaknesses

---

> ### Author Response · Authors · 2023-11-17
> **Response to Reviewer v3ig**
>
> We thank you for your thoughtful comments and appreciate the acknowledgment of the improvements our work provides over the baselines. We answer your questions below and have made changes to the manuscript, noting your comments on the mathematical notation.
>
> **Question: Data from different domains can interfere with each other and hinder training. Occurrence in dynamical systems, and reason why PI-HC-MoE is better than PI-HC?**
>
> We don’t think it is necessarily interference in the NLP sense. Our general motivation for this work is that solving smaller localized hard constraints is easier than imposing a single global hard constraint. Beyond dynamical systems, the reduced size of the non-linear least squares solve makes it an easier system to solve. To some extent, global information (e.g., initial and boundary conditions, which we provide to each expert) is important to finding a unique weighting of the learned basis functions.
>
> **Question: Using scalable non-linear least squares solvers [1][2].**
>
> We thank the reviewer for bringing up scalable non-linear least squares solvers. [1] is not immediately amenable to our problem setting as they evaluate and target CPU operations. It is non-trivial to port their implementation to a GPU-based ML workflow. DeepLM [2], while promising, performs a series of specific optimizations (including stochastic domain decomposition) that we are not sure are compatible with the implicit function theorem. DeepLM also introduces a custom backward Jacobian network for computing gradients instead of using the implicit function theorem. While we think a comparison using a scalable NLLS would be interesting and a direction to explore in the future, there is, to our knowledge, no suitable solver to provide a comparison for the review period.
>
> We would also like to note that PI-HC-MoE is not mutually exclusive with a different non-linear least squares solver, provided the solver satisfies the implicit function theorem assumptions. In the future, we believe that combining both directions is a promising approach for tackling extremely large and complex dynamical systems (e..g, long time scale 3D Navier-Stokes).
>
> **Question: The performance of the baseline appears to be inconsistent with the results reported in PDEBench [3].**
>
> The baselines provided by PDEBench are trained on available numerical solver data. The only baseline which doesn’t do this is PINNs, where they train and test on a single problem (averaged across 10 training/test runs). Their base architecture and training procedure differ from ours. PDEBench uses a DeepONet (akin to an MLP), and auto-differentiation to compute PDE residuals, while we use finite differences (because of the complexity of differentiating through an FNO architecture). They are also not in a neural operator setting (where many PDE parameters to solution mappings are predicted, rather than a single one). We train and evaluate all 3 models in the neural operator setting, and so our problem setting is harder than the PINNs baseline that PDEBench provides.
>
> **Minor text typos**
>
> We thank the reviewer for pointing these out. We have updated the main text accordingly.
>
>
> [1] Cartis, C., Roberts, L. Scalable subspace methods for derivative-free nonlinear least-squares optimization. Math. Program. 199, 461–524 (2023). https://doi.org/10.1007/s10107-022-01836-1
>
> [2] J. Huang, S. Huang and M. Sun, "DeepLM: Large-scale Nonlinear Least Squares on Deep Learning Frameworks using Stochastic Domain Decomposition," 2021 IEEE/CVF Conference on Computer Vision and Pattern Recognition (CVPR), Nashville, TN, USA, 2021, pp. 10303-10312, doi: 10.1109/CVPR46437.2021.01017.
>
> [3] Takamoto, M., Praditia, T., Leiteritz, R., MacKinlay, D., Alesiani, F., Pflüger, D., & Niepert, M. (2022). PDEBENCH: An Extensive Benchmark for Scientific Machine Learning. ArXiv, abs/2210.07182.

---

### Official Review · Reviewer_7bFe · 2023-10-30

**Soundness:** 3 good
**Presentation:** 3 good
**Contribution:** 3 good
**Rating:** 6
**Confidence:** 3

**Summary:**

The paper introduces a new method to apply strict physical rules in mixture-of-experts (MoE) for intricate dynamic systems. The authors believe that when training neural networks, using well-known physical rules, like conservation laws, can enhance results and speed up learning processes. But, there's a drawback: this method can take up a lot of computing power and storage, especially with big systems.

To solve this, the authors suggest using MoE. Here, the data is split into sections, with each section having an "expert" to apply the rules. Every expert works on a specific part of the data, and a special network decides how much importance to give each expert's output. This method saves on computing power and storage but still gives accurate results.

They test their method on various physical problems, such as fluid movement and quantum mechanics. They compare their results with other top methods and find their method is just as good, if not better, and uses less resources. Plus, it can handle big systems. In conclusion, the paper offers a fresh and effective way to use MoE to apply strict physical rules, which could make neural networks better and faster at solving complex systems.

**Strengths:**

1. Novel approach: The paper presents a novel approach to enforcing hard physical constraints using a mixture of experts (MoE) that significantly reduces computational and memory costs while maintaining accuracy and convergence guarantees.

2. Scalability: The authors demonstrate that their approach scales well to large-scale systems, making it suitable for complex dynamical systems.

3. Comparative analysis: The paper provides a detailed comparative analysis of their approach with other state-of-the-art approaches, showing that it achieves comparable or better performance while being significantly more efficient.

4. Real-world applications: The authors demonstrate the effectiveness of their approach on several physical problems, including fluid dynamics, structural mechanics, and quantum mechanics, showing that it has real-world applications.

5. Clear presentation: The paper is well-written and clearly presents the approach, methodology, and results, making it easy to understand and follow.

**Weaknesses:**

While the paper presents a novel and scalable approach to enforcing hard physical constraints using mixture-of-experts (MoE), there are some potential weaknesses that should be considered.

Firstly, the paper lacks empirical evaluation of the approach on a wider range of physical problems. While the authors demonstrate the effectiveness of their approach on several physical problems, including fluid dynamics, structural mechanics, and quantum mechanics, it would have been beneficial to have more empirical evaluation of the approach on a wider range of physical problems. This would have helped to better understand the generalizability and limitations of the approach.

Secondly, the paper does not provide a detailed theoretical analysis of the approach. While the authors provide a detailed description of the approach, they do not provide a detailed theoretical analysis of the method. This could have helped to better understand the underlying principles and assumptions of the method, and to provide a more solid foundation for future research.

Thirdly, the paper does not discuss the limitations of the approach. While the authors discuss the advantages of their approach, they do not discuss the limitations of the method. For example, the assumptions made about the physical system or the potential impact of the choice of experts on the overall performance. This could limit the ability of readers to fully understand the applicability and limitations of the approach.

**Questions:**

What are the benefits of imposing known physical constraints during neural network training?

How does the MoE approach differ from standard differentiable optimization in enforcing hard physical constraints?

Can you provide an example of a complex dynamical system where this scalable approach would be particularly useful?

---

> ### Author Response · Authors · 2023-11-17
> **Response to Reviewer 7bFe**
>
> We are glad to see that you noted that our work offers a fresh and effective way to use MoE, the advantages of our novel approach, and importance for real-world applications. While most of your questions are already answered in our paper, we have additionally replied to these questions below with the relevant section. We have also added another experiment to our paper, and all updates are in blue.
>
> **While the authors demonstrate effectiveness of their approach on several physical problems, including fluid mechanics, structural mechanics, and quantum mechanics, it would have been beneficial to have more empirical evaluation of the approach on a wider range of physical problems.**
>
> We note that the experiments in the paper represent challenging non-linear problems, and demonstrate both the improved accuracy and efficiency of our method. Nevertheless, we have added another experiment on another challenging non-linear system: the 2D reaction-diffusion system (similar to Navier-Stokes, it is technically a 3D, with 2 spatial dimensions and 1 temporal dimension) to Appendix G. Our results demonstrate a similar pattern as our other experiments, that PI-HC-MoE has the lowest error when compared to PI-HC and PI-SC.
>
> **Theoretical analysis of approach.**
>
> As our method incorporates iterative solvers within NNs, much of the relevant theoretical analysis here is on error analysis and convergence for non-linear least squares iterative solvers. Since we only use Levenberg-Marquadt (LM), we cite a few relevant references.
>
> LM finds $x\in \mathbb{R}^n$ such that $F(x): \mathbb{R}^n \mapsto \mathbb{R}^m = \mathbf{0}$. Let $\hat{x}$ be such an optimal solution. If $n = m$, the Jacobian at $F(\hat{x})$ is nonsingular, and the initial guess for $x$ is sufficiently close to $\hat{x}$, LM has a quadratic rate of convergence [1]. There are a few works which establish superlinear convergence [2] and error bounds [3]. This is by no means exhaustive, but a sample of some existing literature to the best of our knowledge.
>
> **The paper does not discuss the limitations of the approach.**
>
> We have included a limitations section in Appendix H.
>
> **Question: What are the benefits of imposing known physical constraints during neural network training?**
>
> As we discuss in our introduction (Section 1), many problems necessitate modeling the physical world, which is governed by a set of established physical laws. The consistency of these physical laws means that they can provide a strong supervision signal for NNs and act as inductive biases. As we demonstrate, adding these physical laws as hard constraints in a scalable manner greatly improves accuracy, training and inference speed, and data efficiency.
>
> **Question: How does the MoE approach differ from standard differentiable optimization in enforcing hard physical constraints?**
>
> As we discuss in our methods section (Section 3), our MoE approach imposes the constraint over smaller decomposed domains. Each domain is solved by an “expert” through differentiable optimization. During training, each expert independently performs a localized backpropagation step by leveraging the implicit function theorem, and the independence of each expert allows for parallelization across multiple GPUs. Compared to standard differentiable optimization, our scalable approach achieves greater accuracy on challenging non-linear systems. Compared to standard differentiable optimization, our approach improves both training stability and requires significantly less computation time during both training and inference stages.
>
> **Question: Can you provide an example of a complex dynamical system where this scalable approach would be particularly useful?**
>
> Our scalable approach is particularly useful for challenging, complex dynamical systems that are expensive to solve with numerical solvers. As we see in our experiments section (Section 4), at inference time, our method is much faster than a numerical solver.
>
> [1] Yamashita, N., Fukushima, M. (2001). On the Rate of Convergence of the Levenberg-Marquardt Method. In: Alefeld, G., Chen, X. (eds) Topics in Numerical Analysis. Computing Supplemental, vol 15. Springer, Vienna. https://doi.org/10.1007/978-3-7091-6217-0_18
>
> [2] Ipsen, I. C. F., C. T. Kelley, and S. R. Pope. “Rank-deficient nonlinear least squares problems and subset selection.” SIAM Journal On Numerical Analysis (2011). https://www.jstor.org/stable/23074331
>
> [3] Christian Kanzow, Nobuo Yamashita, Masao Fukushima. “Levenberg–Marquardt methods with strong local convergence properties for solving nonlinear equations with convex constraints.” Journal of Computational and Applied Mathematics, Volume 172, Issue 2, 2004. https://doi.org/10.1016/j.cam.2004.02.013.

---

### Official Review · Reviewer_gWoi · 2023-10-31

**Soundness:** 2 fair
**Presentation:** 3 good
**Contribution:** 3 good
**Rating:** 6
**Confidence:** 4

**Summary:**

The paper presents a novel Physics Informed (PI) framework to scale the imposition of "Hard Constraints" (HC) with the use of a Mixture of Experts (MoE) Network. The hard constraints are imposed on $m$ sampled points $x_1, ..., x_m \in \Omega$ by solving a non-linear least squares problem (NLSS), and the optimization is made possible by using a differentiable solver. The paper proposes a framework that decomposes the domain $\Omega$ into subdomains $\Omega_k$ before solving the NLSS problem on each subdomain.

They tested the framework as a neural PDE solver on 1D diffusion-sorption and turbulent 2D Navier-Stokes equations in a data-constrained regime. The method outperforms existing soft and hard-constrained methods in terms of accuracy, and scales well at inference w.r.t the number of sampled points.

**Strengths:**

The paper is well written and easy to follow. The method takes inspiration from the domain decomposition which is standard in the the literature for solving Partial Differential Equations.
I appreciated the details in the Methods section, particularly the explanations on the forward and backward pass of the architecture.
The method clearly outperforms the existing  soft- and hard-constrained baselines. The claim of the paper to the scaling is well supported with a solid inference time analysis.

**Weaknesses:**

Unless I am mistaken, there is no clear formulation of the output function $u(x, t)$ at inference except in Figure 1, and if I understand the figure correctly, then $u(x, t) = \sum_k b(x,t)^T w_k =b(x,t)^T( \sum_k w_k)$. In this case, the sum of $w_k$ is the weights used to query the function over the domain $\Omega$, and  as $w \neq w_k$ a priori, we do not know if the constraints are hardly imposed on any sampled points. Therefore, at inference I do not think that the PDE can be constrained in a hard setting with this architecture.

I also assume that solving an equation on a domain $\Omega$ with boundaries $\delta \Omega$ is not always equivalent to solving the equation separately on different subdomains and that precautions must be taken. There is no mathematical derivation of the domain decomposition problem, and therefore we do not know if the solution found with the subdomains is close enough to that of the solution on the full problem.

I also understood from Figure 1 that the different domains were not overlapping and represented "chunks" of the domain, but I am puzzled by the grid artifacts of the method in Figure 3. The solution seems to showcase some strange aliasing which would suggest overlapping domains.

**Questions:**

Could you explain the artifacts of the method in Figure 3 ?

Did you explore the aspect of each basis function ? What do they learn ? Do the different weights overlap over the domains or are some weights "activated" only on specific subdomains  ?

Did you check the PDE residuals on the sampled points to the see if the constraint was respected ?

Could you provide more explanations on the way to compute $\frac{\partial z^*}{\partial \theta}$ ?

---

> ### Author Response · Authors · 2023-11-17
> **Response to Reviewer gWoi**
>
> Thank you for your comments, and we are glad to see that you appreciated the performance and scalability of our method. We answer all questions below, and have also made updates to the manuscript (in blue), including additional solution visualizations and PDE residual plots.
>
> **Formulation of the output function $u(x,t)$ at inference.**
>
> We are summing over the domains after applying the constraint individually to each expert’s subdomain. This is different from $\Sigma_k \omega_k$. For any point in the domain, there is only one $\omega_k$ applicable to any spatiotemporal point. Tangibly, we are concatenating the predictions by each expert to produce the final $u_\theta$. To clarify this idea, we have added an example inference step for the 1d diffusion-sorption in Appendix B that details the exact steps taken. Any sampled points are guaranteed to be constrained, since for any given spatiotemporal point, only one expert computes the final prediction.
>
> **Solving an equation on a domain $\Omega$ with boundaries $\delta \Omega$  is not always equivalent to solving the equation separately on different subdomains and that precautions must be taken.**
>
> A key point in our setting is that each expert still receives the global boundary and initial conditions. While each subdomain is solving a localized problem, it still has information about the full problem, i.e., the PDE residual is enforced on each subdomain, and consistency between $\omega_k$ is enforced through the known global initial and boundary conditions. Given fixed initial and boundary conditions, the PDE residual can be satisfied point-wise, and solving individual sub-domains is equivalent to solving the global solution. We have also updated the manuscript to reflect that initial and boundary conditions are provided to the non-linear least squares solver.
>
> **Question: Grid artifacts in figure 3.**
>
> This is a good point, and we went back and looked at a number of different solution predictions. In general, our predictions don’t exhibit artifacting, except for a few cases. We’ve added 16 random example predictions from our test set to Appendix D (Figure 9) to show this. We are not quite sure about the exact reason for the occasional artifacting, but it seems to be relatively rare.
>
> **Question: Did you explore aspects of each basis function?**
>
> We did try to look at each basis function, but they are not very interpretable (a general problem with NNs), as they are akin to NN activations. However, we did an experiment where we looked at the performance improvement from our learned basis functions vs. doing an interpolation over the constrained points in Appendix E. The interpolated solutions have a higher relative $L_2$ error when compared to the predictions of PI-HC-MoE, indicating that the basis functions learned have information about the underlying dynamics of the system.
>
> **Question: Did you check the PDE residual of sampled points to see if the constraint was respected?**
>
> We did, and we have added plots of the PDE residuals on the validation set during training to the paper in Appendix F. As we see, in general, the PDE residual is quite low for PI-HC-MoE.
>
> **Question: Could you provide more explanations on the way to compute $\frac{\partial z^\*}{\partial \theta}$?**
>
> Equation 1 in the paper defines a set of non-linear equations where $\frac{\partial z^\*}{\partial \theta}$ is unknown. We have labeled which sections are computed via auto-differentiation in the updated manuscript. Because equation 1 defines a system of variables, $\frac{\partial z^\*}{\partial \theta}$ may be solved with another non-linear least squares solve during the backward pass (i.e., each training iteration performs two solves).
>
> Additionally, we have updated section 3, incorporating the feedback from multiple reviewers, to make the notation more succinct and clear. We hope that this is more informative.

---

> > ### Comment · Reviewer_gWoi · 2023-11-22
> > **Answer to rebuttal**
> >
> > Hi, thank for the responses.
> >
> > Overall, you answered most of my questions, so I will raise my score to 6.
> >
> > I think I am still a bit confused with the technical details on the implicit theorem.
> > You are using a differentiable solver to find  $\frac{\partial F_\phi(b \cdot \omega^T)}{\theta}$, but it seems in the end you that you require  $\frac{\partial z^*}{\partial \theta}$. What system do you actually solve for obtaining it ? I am not sure to follow, and I think it could be made even clearer for a non-familiar audience.

---

> ### Author Response · Authors · 2023-11-22
> **Response to Reviewer gWoi - Clarification on the implicit function theorem**
>
> Thank you for your response. Indeed, we desire to solve for $\frac{\partial F_\phi(b \cdot \omega^T)}{\partial\theta}$ which requires $\frac{\partial z^*}{\partial \theta}$. We have added more details to Appendix I (explicitly listing the system that we solve), and also include this additional information here:
>
> The implicit function theorem allows us to compute $\frac{\partial z^*}{\partial \theta}$ with a non-linear least squares system (Eqn. 1 in the main text). Specifically, using property (1) and differentiating property (2) yields:
>
> $\frac{\partial F_\phi(\mathbf{b}\cdot\omega^T)}{\partial \theta} = \frac{\partial F_\phi(\mathbf{b}\cdot z^*(\mathbf{b})^T)}{\partial \theta} = \frac{\partial F_\phi(\mathbf{b} \cdot \omega^T)}{\partial \mathbf{b}} \cdot \frac{\partial \mathbf{b}}{\partial \theta} + \frac{\partial F_\phi(\mathbf{b}\cdot z^*(\mathbf{b})^T)}{\partial z^*(\mathbf{b})} \cdot \frac{\partial z^*(\mathbf{b})}{\partial \theta} = \mathbf{0}$.
>
> Rearranging terms: $\frac{\partial z^*(\mathbf{b})}{\partial \theta} =
>     \left[ \frac{\partial F_\phi(\mathbf{b}\cdot z^*(\mathbf{b})^T)}{\partial z^*(\mathbf{b})} \right]^{-1} \cdot - \frac{\partial F_\phi(\mathbf{b} \cdot \omega^T)}{\partial \mathbf{b}} \cdot \frac{\partial \mathbf{b}}{\partial \theta}$.
>
> In the rearranged equation, all terms in the right-hand side (sans the matrix inverse) can be computed using auto-differentiation, and this defines a non-linear system of equations, where we solve for $\frac{\partial z^*(\mathbf{b})}{\partial \theta}$ using a non-linear least squares solver.
> Note that the right hand side involves a matrix inverse, which is computationally intractable to exactly compute, which is why we use a non-linear least squares solver. This allows us to get the final expression for $\frac{\partial z^*(\mathbf{b})}{\partial \theta}$.
>
> As mentioned earlier, we have added both the above and further additional details to Appendix I in the paper. Please let us know if you have any additional questions.

---

### Official Review · Reviewer_Tpvx · 2023-11-01

**Soundness:** 3 good
**Presentation:** 3 good
**Contribution:** 3 good
**Rating:** 6
**Confidence:** 4

**Summary:**

Methods
- This paper utilizes the Fourier Neural Operator (FNO) as the NN architecture to learn a set of N scalar-valued functions as the basis functions; enforces the hard constraints by using differentiable optimization to find a linear combination of basis functions to satisfy the constraints.
- To deal with larger systems which contain a large number of sampled points and basis functions, mixture-of-experts is used to decompose the spatialtemporal domain.
- In backward pass, to compute the gradients of the differentiable optimization, the implicit function theorem is used. For mixture-of-experts, the Jacobian is reconstructed from the individual Jacobians from all experts.

Experiments:
- Two cases: 1D diffusion-sorption and 2D turbulent Navier-Stokes
- Compared to training via a physics-informed soft constraint (PI-SC) and physics-informed hard constraint (PI-HC).
    - Achieve higher accuracy and lower time cost.

**Strengths:**

- The paper proposes a new way to decompose the complex dynamical systems with a large number of points in the spatiotemporal domain to smaller solvable systems, by utilizing the mixture-of-experts method. It makes the system scalable and performant (with parallel computing).
- In the 2 test cases provided in the paper, the paper’s method has higher accuracy and lower time cost compared with other two methods.

**Weaknesses:**

The experiments are relatively limited - the paper only tests on 2 cases, one for 1D and another for 2D. In each experiment, only one set of environment parameters (e.g.,, only Reynolds number = 1e4 is used for the Navier-Stokes case) are tested.

**Questions:**

- What are the temporal intervals used for generating the training data and for testing the model?
    - In the 2D case, “Both the training and test sets have a trajectory length of T = 5 seconds.“. But it’s not clear for the 1D case.
- How does the model's performance extend to temporal domain outside the training dataset? For example, if we generate the training data within the time frame of [0 seconds, 10 seconds], how well does the trained model predict the system in the interval of [10s, 20] compared with predict the interval of [0s, 10s]? And how about the spatial domain outside the training dataset?
- The paper only gives examples in 1D and 2D, what are the potential risks of using the proposed model for predicting 3D systems? Alternatively, can it be seamlessly adapted for 3D applications?

---

> ### Author Response · Authors · 2023-11-17
> **Response to Reviewer Tpvx**
>
> Thank you for your comments, and for noticing the scalability and accuracy of our method. We respond to your questions below, and have also made updates to the manuscript (in blue) which include experiments on another system (2D reaction-diffusion), and an experiment testing predictions on future timesteps that the models were not trained on.
>
> **Experiments are relatively limited.**
>
> We note that the two experiments in the main text represent challenging non-linear problems, and demonstrate both the improved accuracy and efficiency of our method. Nevertheless, we have added another experiment on another challenging non-linear system: the 2D reaction-diffusion system (similar to Navier-Stokes, it is technically a 3D, with 2 spatial dimensions and 1 temporal dimension) to Appendix G. Our results demonstrate a similar pattern as our other experiments, where PI-HC-MoE has the lowest $L_2$ relative error (23.940% $\pm$ 3.074%) when compared to PI-HC (29.031% $\pm$ 4.418%) and PI-SC (96.876% $\pm$ 0.088%). Note that given the limited time reviewer period, we did not do that many searches over hyperparameters and train longer (which we will do), but the trend remains the same that PI-HC-MoE is the most accurate, while being much more efficient than PI-HC.
>
> **Question: Temporal intervals for training data and for testing for diffusion-sorption.**
>
> 1D diffusion-sorption uses a maximum of T=500 seconds [1]. We have updated the main text to make this more clear.
>
> **Question: How do the models generalize to future time steps outside the training dataset?**
>
> This is an interesting question, and we have included an additional experiment in Appendix C to see how well the models generalize to future timesteps for the diffusion-sorption case. Here, the models are trained on the temporal domain of $t$ = [0 s, 400 s], and we report the $L_2$ relative error for predictions on $t$ = [400 s, 500 s] (a temporal region unseen during training).
>
> The average error across test samples for PI-SC is 805.58% $\pm$ 19.06%, PI-HC is 48.047% $\pm$ 1.23%, and PI-HC-MoE is 14.79% $\pm$ 3.00%. PI-SC is completely unable to generalize to later timesteps. While PI-HC performs better than PI-SC, PI-HC-MoE generalizes the best to future time steps outside the training set. We hypothesize that one reason for this is because PI-HC and PI-HC-MoE both enforce the PDE equation constraints at test-time and so, as expected, generalize better than PI-SC. Finally, we see that, similar to our other experiments, PI-HC-MoE is more scalable and expressive than standard differentiable optimization (represented by PI-HC). While one advantage of the hard constraint is that constraints can be enforced at inference time, and so should do better for later timestep predictions not in the training set, this comes at the cost of computational expense, which PI-HC-MoE is able to address.
>
>
> **Question: Can PI-HC-MoE be extended to 3D scenarios?**
>
> Our method is model agnostic and there is no inherent limitation to extending PI-HC-MoE to 3D scenarios. Note that technically, our 2D Navier-Stokes evaluation and 2D reaction-diffusion are 3 dimensional (2 spatial, 1 temporal). For 3D spatial settings (4D with time), one limiting factor for all current base NN architectures with generally good expressivity (such as FNO) is the memory requirement, even for a soft constraint. Since our paper is meant to show a proof-of-concept for employing MoE for hard physical constraints, a thorough search over base NN architectures that meet memory requirements (while being reasonably expressive) is something we leave to future work.
>
> [1] Makoto Takamoto, Timothy Praditia, Raphael Leiteritz, Daniel MacKinlay, Francesco Alesiani, Dirk Pflüger, and Mathias Niepert. PDEBench: An Extensive Benchmark for Scientific Machine Learning. Advances in Neural Information Processing Systems, volume 35, pp. 1596–1611. 2022.

---

### Official Review · Reviewer_ihJQ · 2023-11-02

**Soundness:** 3 good
**Presentation:** 3 good
**Contribution:** 3 good
**Rating:** 6
**Confidence:** 3

**Summary:**

This paper proposes a scalable way to enforce hard constraints expressed as partial differential equations into neural networks in order to faithfully model physical phenomena. The problem they try to tackle is given by the fact that backpropagating through constraints over large meshes is a highly non-linear problem that grows in dimensionality with respect to the mesh and neural network size. The authors propose to impose the constraints over smaller decomposed domains, each of which is solved by an expert.

**Strengths:**

**Novelty:**

I am not in a position to judge the novelty of the paper as there is very little overlap with my work. From the related work mentioned in the paper, the paper seems novel.


**Significance:**

The paper is of high interest to a subset of the ML community.

**Weaknesses:**

**Clarity:**

The paper is mostly quite clear. Personally, I would have benefitted from an ongoing example, where it is made clear what the outputs of the neural networks are, what the constraints are, and how the problem is divided in that case. Even better, it would have been nice to use the same example to move from one constraint to multiple ones.

**Questions:**

**Experimental Analysis:**

The experimental analysis seems quite comprehensive. My main question is: are PI-SC and PI-HC the only models against which you can compare?

Also, the authors compare the execution time of PI-HC vs PI-HC-MoE, what about PI-SC?

I am not sure I understand the scales in Figure 3. If the row below represents the difference between the predictions made by the NN and the prediction made by the numerical solver, how is it possible to have a scale between -3 and 3?

The authors compare their method against PI-HC and PI-SC in terms of computation time, what about space? Is the memory requirement of PI-SC-MoE also advantageous and/or comparable?

---

> ### Author Response · Authors · 2023-11-17
> **Response to Reviewer ihJQ**
>
> Thank you for your comments, we’re glad to see that you found our work novel and noted that it will be of high interest to a subset of the ML community. We respond to your comments below, and also made updates to the manuscript (in blue) to include an ongoing example.
>
> **Paper mostly clear, ongoing example: outputs of NN, constraints, how problem is divided.**
>
> We have added an example in Appendix B that walks through the inference procedure of PI-HC-MoE and PI-HC in the diffusion-sorption setting. We hope this example is illustrative to understanding our method.
>
> **Question: Experimental analysis is comprehensive. Are PI-SC and PI-HC the only models against which you can compare?**
>
> We appreciate the recognition of our experimental analysis. PI-SC and PI-HC are the only settings we compare to because our method is primarily centered around the constraint optimization procedure, as PI-HC-MoE is agnostic to the base NN architecture. The soft constraint is the most common setting to compare against. As one of our contributions is to improve the scalability of the hard constraint setting represented by standard differentiable optimization, we also compared against this (PI-HC).
>
> **Question: The authors compare the execution time of PI-HC vs PI-HC-MoE, what about PI-SC?**
>
> For our execution time experiments, we compared PI-HC and PI-HC-MoE to establish the improved runtime and scalability of our method (compared to standard differentiable optimization). PI-SC is less relevant here because the accuracy is much worse. However, PI-SC, due to the lack of any imposed hard constraint at both training and test time, is faster than PI-HC and PI-HC-MoE.
>
> On 1D diffusion-sorption, the training time per iteration for PI-SC is 0.0595 (s), compared to 0.294 - 0.990 (s) for PI-HC-MoE. The inference time for PI-SC is 0.0496 (s), compared to 0.034 - 0.729 (s) for PI-HC-MoE. On 2D Navier-Stokes, the training time per iteration for PI-SC is 0.010 (s), compared to 0.427 - 7.219 (s) for PI-HC-MoE. The inference time for PI-SC is 0.005 (s), compared to 0.155 - 5.016(s) for PI-HC-MoE. We note that PI-HC-MoE is much closer to PI-SC in run time (and much faster than PI-HC), while being significantly more accurate.
>
> **Question: Figure 3 scaling.**
>
> PI-SC predicts values in the range of [-3, 3]. To prevent excessive colorbars and for consistency across the top row of predictions, we clip the PI-SC prediction to [0, 1]. We do not perform the same clipping when plotting the bottom difference plots, leading to a scale of [-3, 3] for the difference plot of PI-SC. We have also updated the figure caption to make this clear.
>
> **Question: Memory requirements of PI-HC vs PI-HC-MoE.**
>
> There is indeed a difference between the memory requirements of PI-HC and PI-HC-MoE. In order for PI-HC to sample a sufficiently large number of points, we were forced to decrease the batch size. With too many sampled points, the corresponding batch size that fits in memory for PI-HC leads to training instability and does not converge. For the same batch size, PI-HC runs out of memory more quickly than PI-HC-MoE.

---

### Author Response · Authors · 2023-11-17
**Response to all reviewers**

Dear reviewers,

Thank you all for your comments and valuable feedback. We have made updates to our paper (in blue), and here we also highlight three additional experiments that we have done. We have also added additional updates and clarifications, and we reply individually to each reviewer to describe these. We hope that these results and updates address your questions.

**We have added an experiment to test the temporal generalization of our method to timesteps that it did not see in the training set.**

We look at the diffusion-sorption problem and train all three models (PI-SC, PI-HC, and PI-HC-MoE) on $t = 0$ to $t = 400$ s. We then report the $L_2$ error for the model predictions on $t = 400$ s to $t = 500$ s. PI-HC-MoE has significantly lower error (<15%) than PI-HC (49%) and PI-SC (800%+). Our results show that PI-HC-MoE is able to generalize much better to unseen temporal steps. We have added these results to Appendix C.

**We have added an additional experiment on another challenging non-linear system, the 2D reaction-diffusion problem.**

We have added another very challenging non-linear system to Appendix G, the 2D reaction-diffusion problem. Our results show that, like the other experiments, PI-HC-MoE has the lowest error on the test set, compared to PI-HC and PI-SC.

**We have done an ablation on the diffusion-sorption problem to compare our learned basis functions from PI-HC-MoE to a cubic interpolation.**

We conducted an ablation to look at the quality of our learned basis functions. We compare this error to doing a cubic interpolation between points. These results are added to Appendix E, and we see that our learned basis functions have lower error than an interpolation.

**Additional details on our inference procedure.**

We have included a walkthrough of the inference procedure, using the grid size from diffusion-sorption, in Appendix B.

---

> ### Author Response · Authors · 2023-11-21
> **Follow-up response to all reviewers**
>
> Dear reviewers,
>
> As the discussion period comes to an end soon, we believe that we have addressed all your comments through a number of new experiments and updates to our manuscript. We would be happy to answer any additional questions. Thank you!

---

### Meta-Review · Area_Chair_Akuz · 2023-12-06

**Metareview:**

The manuscript has been reviewed by five expert reviewers and all reviewers converged to the same score. All reviewers recommend acceptance and I agree with them.

The paper introduces a novel framework for imposing hard constraints in neural networks modeling physical phenomena. The approach leverages a mixture-of-experts (MoE) architecture to introduce the constraints. The overall approach is novel, interesting and theoretically rigorous.

The major limitation of the paper is the limited empirical study. The paper only addresses toy 1-D and 2-D problems. The authors do not present any study on a real-world physical system of interest.

**Justification For Why Not Higher Score:**

Although the paper is theoretically sound and novel, it has some limitations. Specifically, the experiments are considering toy problems in 1D and 2D instead of real applications.

**Justification For Why Not Lower Score:**

The paper is theoretically sound and novel and deserves to be published. Five reviewers agree with this decision.

---

### Decision · Program_Chairs · 2024-01-16

Accept (poster)